# The relationship between consultation modality and mental health professional type and perceived quality of care and mental health outcomes: A comparison between employed and unemployed individuals

Aishvinigaa Sathananthan[1], Kishana Balakrishnar[1], Raihana Premji[1], Behdin Nowrouzi-Kia[1,2,3,4]*

**1** Department of Occupational Science and Occupational Therapy, Temerty Faculty of Medicine, University of Toronto, Toronto, Ontario, Canada, **2** Krembil Research Institute-University Health Network, Toronto, Ontario, Canada, **3** Centre for Research in Occupational Safety & Health, Laurentian University, Sudbury, Ontario, Canada, **4** Institute for Mental Health Policy Research, Center for Addiction and Mental Health, Toronto, Ontario, Canada

* behdin.nowrouzi.kia@utoronto.ca

## Abstract

Mental health is becoming increasingly acknowledged as a critical factor to one's well-being. Poor mental health can have serious impacts, particularly among working and non-working adults, requiring the need for effective treatments and interventions tailored to them. The type of mental health professional consulted and the type of consultation modality may be associated with individuals' perceptions of the quality of care that they receive as well as their mental health outcomes. Therefore, this study aims to examine the relationship between the type of consultation with different mental health professionals and the perceived quality of care and mental health outcomes among employed and unemployed Canadians. Data was utilized from the 2024 Mental Health and Access to Care Survey by Statistics Canada. Independent variables included the type of mental health professional consulted (psychiatrist, family doctor, psychologist, and social worker) and consultation modality (in-person, remote, and hybrid). Primary outcome variables were self-reported quality of care and perceived mental health outcomes. Multivariable logistic regression models were used to calculate odds ratios and 95% confidence intervals, controlling for age, gender, marital status, and visible minority status. A total of 1,222 participants were analyzed. Unemployed participants who accessed hybrid psychiatry services were significantly less likely, about one-sixth, to report lower quality of care than those using in-person services (AOR: 0.16; 95% CI: 0.02-0.98). Similarly, those who used hybrid family doctor services were less likely, about one-fifth, to report lower quality of care (AOR: 0.21; 95% CI: 0.07-0.66). No statistically significant differences in mental health outcomes were found across consultation modality and professional type for

**Data availability statement:** The dataset used and/or analyzed during the current study is publicly available through the Odesi database. Please see https://odesi.ca/en/details?id=/odesi/doi__10-5683_SP3_ABG9CY.xml.

**Funding:** The authors received no specific funding for this work.

**Competing interests:** The authors have declared that no competing interests exist.

both employed and unemployed groups. By analyzing both the modality of care and the professional consulted, notable differences emerged in self-reported quality of care. Future research should further explore how inter-professional collaboration can improve the quality of care and mental health outcomes.

## Introduction

### Background

Recognition of mental health as a critical factor to one's overall well-being has been on the rise as society as a collective has begun to prioritize mental health due to its multifaceted impact on our daily lives [1]. There is a wide array of factors that can contribute to the detriment of one's mental health, some of which include poor working and living conditions, personal and familial stressors and traumatic events [1,2]. Similarly, poor mental health conditions can have adverse effects on one's personal and work lives. Mental health has become a prominent issue on a global scale, with an estimated 15% of adults of working age having a mental disorder in 2019, potentially leading to adverse financial impacts and productivity loss [3]. To address these concerns, consultations with mental health professionals, such as psychiatrists, family doctors and psychologists, play an essential role in providing diagnosis, treatment, and support. However, the way these consultations are delivered, as well as whoever offers them, can significantly impact the quality, accessibility, and consistency of care patients receive, ultimately influencing their mental health outcomes.

While many mental health professionals have been providing care in-person for years, recent changes due to the COVID-19 pandemic have led to a significant shift towards a technological approach to healthcare, known as telehealth [4]. In mental health, this typically refers to live therapeutic sessions between the clinician and the patient, conducted via telephone or video conference [5]. Although telehealth services existed before the pandemic, they were not widely utilized [5]. However, at the start of the pandemic, many countries reduced their in-person services, leading to a sharp increase in the use of telemental health [6]. Within a year, both general and specialized care had become almost completely virtual [6]. For instance, in the United States, approximately 88.1% of mental health treatment facilities offered telehealth services in September 2022, a significant increase from 39.4% in April 2019, indicating the expansion of telehealth availability for mental health services and care due to state-level policy changes during that period [7]. This diversity in service provision offers numerous benefits, including greater accessibility for individuals in rural areas, reduced costs, and fewer barriers for those with disabilities [8–10]. Additionally, this new method of offering mental health services has been revolutionary, with many choosing to continue offering services over various modalities, even after in-person services resumed [5].

Given the recent emergence of telehealth services, it is crucial to determine whether the modality of service provision and the type of mental health professional have a significant impact on self-reported quality of care and perceived mental health

outcomes. A systematic review in the literature compared the efficacy of telehealth services and in-person services for improving mental health outcomes [11]. The results of the review found that mental health outcomes were significantly improved when the intervention was delivered by a psychologist or psychiatrist via telehealth strategies as compared to face-to-face interventions [11]. Similarly, telehealth appears to be particularly beneficial for social workers who care for vulnerable populations.

The use of technology to provide support and therapeutic interventions reduces many systemic barriers that may have otherwise prevented access, including transportation, geographic limitations, and socioeconomic difficulties [12]. Moreover, a systematic review highlighted positive patient experiences regarding remote consultation [13]. For instance, 66.4% of patients receiving mental health care reported satisfaction with remote consultation; however 86.2% were more satisfied with in-person care. Remote consultations were associated with reduced anxiety, often due to shorter wait times or increased flexibility [13]. Some patients also felt more comfortable communicating through textual and asynchronous methods, allowing them to express themselves more clearly. However, three studies from this systematic review reported negative patient experiences, with some finding remote consultations to be more time-consuming. In contrast, patient satisfaction was found to be comparable between telehealth and in-person therapy services in another study [14]. However, there were some reports of discomfort regarding online group sessions [14]. This suggests that while mental health outcomes and perceived quality of care appear to be consistent or improved through telehealth, some forms of intervention might be better suited for private, in-person sessions, having implications for therapists who offer a greater variety of services, including group therapy.

In addition to consultation modality and type of mental health professional, individual factors such as employment status may also influence how people perceive mental health outcomes. Employed and unemployed individuals may perceive quality of care and mental health differently. One study, for example, found that unemployed individuals, compared to those with full-time employment, were more likely to report lower self-perceived mental health [15]. Similarly, another study found a positive link between employment and mental health for both women and men [16].

Previous studies in the literature have examined consultation modality, type of mental health professional, and employment status, either individually [17] or in combination with two of these factors [18]. One study, for example, investigated whether employment status influenced the type of professional consulted (general practitioner or psychiatrist); however, it was analyzed in terms of countries with high mean unemployment rates compared with those with lower rates, rather than individual employment status [16]. To our knowledge, no single study has examined the combined effects of these two factors and explored how they interact with individual characteristics, such as employment status. This highlights a gap in the literature requiring the need to examine the interactions between mode of consultation, mental health professional, and employment status, which is essential as it can inform mental health services by highlighting which types of mental health care and delivery methods support better outcomes for employed and unemployed individuals. The results from our work would aid in the development of effective targeted interventions for these individuals based on their employment status.

## Theoretical framework

This study is guided by both the Donabedian's Quality of Care Framework and Andersen's Behavioural Model of Health Services Use, allowing for a more comprehensive lens for examining differences in mental health care experiences. Donabedian's framework evaluates the quality of healthcare across three domains: structure, process and outcome, where structure is defined as the aspects of a setting where care occurs, process is how care is delivered, and outcome is the effects of that care [19]. This is a very relevant and important guide to our study as it helps us understand that the type of mental health professional providing care represents the structure domain, the type of consultation modality reflects the process domain, and perceived quality of care and mental health outcomes are part of the outcome domain. Andersen's Behavioural Model of Health Services Use adds on to this perspective by highlighting the role of predisposing factors, enabling resources and need in shaping how health services are used and their outcomes [20]. In the context of this study, employment status is

seen as both a predisposing characteristic and an enabling resource that influences access to care and service utilization. Therefore, combining these two frameworks provides a strong foundation for the study, helping to establish patterns in the perceived quality of care and mental health outcomes, while also offering potential explanations for the findings.

## Objectives

This study aims to examine the relationship between the type of consultation (in-person, remote, or hybrid) with different types of mental health professionals (e.g., psychiatrists, psychologists, family doctors, and social workers) and the quality of care and self-reported mental health among employed and unemployed Canadians. In the context of this study, family doctors are considered as mental health professionals alongside psychiatrists and psychologists, due to their role in assessing, treating and referring patients for mental health concerns [21–23].

## Methods

### Ethics statement

Ethical approval was not required for this study at our institution as the data used is already publicly available. Please see https://research.utoronto.ca/ethics-human-research/activities-exempt-human-ethics-review.

### Dataset

This study employed a cross-sectional, secondary data analysis of the 2024 Mental Health and Access to Care Survey (MHACS) dataset from Statistics Canada to address the research question [24]. Secondary data analysis is a well-utilized approach that can examine population-level trends through the use of large representative datasets [25,26]. This dataset is publicly available and was accessed from the Odesi database on April 7, 2025 (https://odesi.ca/en/details?id=/odesi/doi__10-5683_SP3_ABG9CY.xml) [27]. Additionally, the authors had no access to identifiable information regarding individual participants at any time. The survey collected detailed information about the mental health status of Canadians, as well as participants' access to and need for services and supports, making this dataset appropriate for this study. It has also been utilized in several studies in literature demonstrating its value as a data source for mental health research [28–30]. Furthermore, the MHACS examines how the COVID-19 pandemic affects population health and evaluates changes in patterns of mental health, service utilization, and functioning over the past decade. In this context, functioning refers to the ability to carry out daily activities. Data collection commenced on March 17, 2022, and ended on July 31, 2022. The MHACS employed a stratified simple random sampling method. The survey was structured to oversample individuals identifying with the four largest population groups in Canada, designated as visible minorities (Black, Chinese, Filipino, South Asian, and others, which encompasses the rest of the analytic population). This approach uses a 0.5 power allocation to ensure proportional representation based on the relative size of each group.

### Dependent variables

The two dependent variables include perceived quality of care and perceived mental health outcomes. Regarding the quality of care, the survey asked participants to rate the level of help they received from the healthcare professional they sought. Participants had the option to rate it as "A Lot", "Some", "a Little", and "Not at All". Responses that indicated "Don't Know" were excluded from the analysis. For measuring perceived mental health, participants were asked in the survey to rate their condition using a 5-point Likert scale with options including "Excellent", "Very Good", "Good", "Fair", and "Poor".

### Independent variables

The independent variables in the study are the type of mental health professional consulted (psychiatrist, family doctor, psychologist, and social worker) and the modality of consultation (in-person, over the phone, video, and text message).

The original dataset included 16 binary variables based on survey questions that asked whether participants had sought help from each professional-modality combination within the past 12 months. For each question, participants, if applicable, indicated "yes" or "no" to denote if they received the specific support method. Consultation modalities were recoded into three broader categories for each healthcare professional (in-person only, remote only, and hybrid), yielding three new categorical variables for each provider type.

### Statistical analyses

All statistical analyses were performed using R version 4.5.0 on MacOS. Descriptive statistics were conducted to identify the frequencies of demographic variables, including age, gender, marital status, and visible minority status, as well as the frequencies of professional-modality combinations pursued. Multivariable logistic regression models were used to generate odds ratios and 95% confidence intervals (CIs) for predicting poor quality of care and mental health, based on a series of independent variables. To facilitate analysis, the two dependent variables were dichotomized. For quality of care, responses "a lot" and "some" were combined to indicate an adequate level of care received from their support delivery, whereas responses "a little" and "not at all" were combined to signify the poor level of care. For perceived mental health, responses "Excellent", "Very Good", and "Good" were combined to represent high self-perceived mental health, whereas "Fair" and "Poor" were combined to specify poor mental health. Analyses treated quality of care and perceived mental health as dichotomous outcomes to enhance interpretability to relevant stakeholders.

Co-variants were added in the model to adjust for differences across demographic variables including age, gender, marital status, and visible minority status. Employment status acted as a moderator variable to identify the varying relationships across these groups. Statistical differences were deemed significant at the 5% level. The variance inflation factor (VIF) was used to detect any presence of multicollinearity.

### Results

A total of 1,122 participants were included in the analysis. Table 1 presents a breakdown of the key demographic characteristics between the employed and unemployed participants. The majority of employed participants in the study (38.48%) are between the ages of 35 and 54 years old, and most identify as women+ (64.94%). Additionally, a greater proportion of employed individuals are unmarried (69.87%) and do not identify as a visible minority (63.16%). In contrast to the employed individuals, many of the unemployed participants (45.78%) are younger, primarily between the ages of 18 and 24. Similar to the employed group, however, most unemployed participants identify as women+ (66.57%), are unmarried (76.81%), and do not identify as a visible minority (57.23%).

Table 2 compares the frequencies of consultation types across different healthcare professions between employed and unemployed individuals. The choice of healthcare provider and consultation type was consistent among the different employment status groups. A greater proportion of employed (60.51%) and unemployed participants (60.24%) sought support from a family doctor compared to other healthcare providers. Among those who accessed psychiatric services, remote consultations were the most common for both employed individuals (11.27%) and unemployed individuals (13.86%). For participants who consulted a family doctor, in-person services were the most frequently reported, by 25.32% of employed individuals and 25.30% of unemployed individuals. Additionally, most individuals who consulted a psychologist did so remotely for both employed (16.20%) and unemployed participants (16.57%). Similarly, among those who received help from a social worker, the majority were assisted through remote services for both employed individuals (26.20%) and those who were unemployed (24.40%). Hybrid services were the least utilized services across all healthcare professionals for both groups.

Table 3 presents the adjusted odds ratios for predicting poor quality of care received by healthcare consultation types across the different employment groups. Among unemployed individuals, those who sought hybrid psychiatry services were significantly less likely or about one-sixth as likely to report lower quality of care than those who sought in-person

**Table 1. Demographic characteristics of study sample (n = 1,122).**

| Variables | Employed Individuals (n = 790) | | Unemployed Individuals (n = 332) | |
| --- | --- | --- | --- | --- |
| | Frequency (n) | Percentage (%) | Frequency (n) | Percentage (%) |
| **Age Group** | | | | |
| 15-24 Years Old | 220 | 27.85% | 152 | 45.78% |
| 25-34 Years Old | 184 | 23.29% | 41 | 12.35% |
| 35-54 Years Old | 304 | 38.48% | 86 | 25.90% |
| 55+ Years Old | 82 | 10.38% | 53 | 15.96% |
| **Gender** | | | | |
| Men+† | 277 | 35.06% | 111 | 33.43% |
| Women+‡ | 513 | 64.94% | 221 | 66.57% |
| **Marital Status** | | | | |
| Married | 238 | 30.13% | 77 | 23.19% |
| Unmarried § | 552 | 69.87% | 255 | 76.81% |
| No | 499 | 63.16% | 190 | 57.23% |
| Yes | 291 | 36.83% | 142 | 42.77% |

† Includes men, boys, and some non-binary persons.

‡ Includes women, girls, and some non-binary persons.

§ Encompasses those who were never married, widowed, separated or divorced, and living common law.

**Table 2. Frequencies of consultation types across different healthcare professions among employed and unemployed individuals (n = 1122).**

| Variables | Employed Individuals (n = 790) | | Unemployed Individuals (n = 332) | |
| --- | --- | --- | --- | --- |
| | Frequency (n) | Percentage (%) | Frequency (n) | Percentage (%) |
| Psychiatrist | | | | |
| In-Person | 35 | 4.43% | 22 | 6.63% |
| Remote | 89 | 11.27% | 46 | 13.86% |
| Both | 27 | 3.42% | 14 | 4.22% |
| Total | 151 | 19.11% | 82 | 24.70% |
| Family Doctor | | | | |
| In-Person | 200 | 25.32% | 84 | 25.30% |
| Remote | 196 | 24.81% | 77 | 23.19% |
| Both | 82 | 10.38% | 39 | 11.75% |
| Total | 478 | 60.51% | 200 | 60.24% |
| Psychologist | | | | |
| In-Person | 63 | 7.97% | 16 | 4.82% |
| Remote | 128 | 16.20% | 55 | 16.57% |
| Both | 31 | 3.92% | 9 | 2.71% |
| Total | 222 | 28.10% | 80 | 24.10% |
| Social Worker | | | | |
| In-Person | 70 | 8.86% | 55 | 16.57% |
| Remote | 207 | 26.20% | 81 | 24.40% |
| Both | 48 | 6.08% | 17 | 5.12% |
| Total | 325 | 41.14% | 153 | 46.08% |

services (AOR: 0.16; 95% CI: 0.02-0.98). However, this relationship was insignificant for employed individuals. Moreover, unemployed participants who accessed hybrid family doctor services were significantly less likely or about one-fifth as likely to report lower quality of care received compared to those who used in-person services (AOR: 0.21; 95% CI: 0.07-0.66). Likewise, this association did not reach statistical significance for employed participants. Among social workers, the quality of care did not differ significantly across modalities between the two employment groups. Table 4 shows the adjusted odds ratios and their 95% confidence intervals for poor mental health outcomes by healthcare consultation type across both employment groups. No statistically significant differences were observed in mental health outcomes across

**Table 3. Adjusted odds ratios for poor quality of care received by healthcare consultation type among employed and unemployed individuals.**

| Healthcare Professional | Consultation Type | Employed Individuals | | Unemployed Individuals | |
|---|---|---|---|---|---|
| | | AOR^ | 95% CI | AOR^ | 95% CI |
| Psychiatrist | In-Person | 1.00 (ref) | N/A | 1.00 (ref) | N/A |
| | Remote | 2.17 | 0.84-5.65 | 0.39 | 0.10-1.44 |
| | Hybrid | 0.71 | 0.18-2.77 | 0.16 * | 0.02-0.98 |
| Family Doctor | In-Person | 1.00 (ref) | N/A | 1.00 (ref) | N/A |
| | Remote | 0.88 | 0.57-1.36 | 1.19 | 0.61-2.33 |
| | Hybrid | 0.65 | 0.36-1.18 | 0.21 * | 0.07-0.66 |
| Psychologist | In-Person | 1.00 (ref) | N/A | 1.00 (ref) | N/A |
| | Remote | 1.38 | 0.60-3.20 | 0.38 | 0.09-1.50 |
| | Hybrid | 0.38 | 0.08-1.89 | 0.17 | 0.02-1.65 |
| Social Worker | In-Person | 1.00 (ref) | N/A | 1.00 (ref) | N/A |
| | Remote | 1.02 | 0.53-1.98 | 1.34 | 0.52-3.50 |
| | Hybrid | 0.50 | 0.18-1.40 | 0.70 | 0.12-4.03 |

* Significant at the a = 0.05 level.

^ Adjusted for Age, Gender, Visible Minority Status, and Marital Status.

**Table 4. Adjusted odds ratios for poor mental health by healthcare consultation type among employed and unemployed individuals.**

| Healthcare Professional | Consultation Type | Employed Individuals | | Unemployed Individuals | |
|---|---|---|---|---|---|
| | | AOR^ | 95% CI | AOR^ | 95% CI |
| Psychiatrist | In-Person | 1.00 (ref) | N/A | 1.00 (ref) | N/A |
| | Remote | 1.03 | 0.44-2.42 | 2.85 | 0.78-10.45 |
| | Hybrid | 1.66 | 0.54-5.08 | 2.46 | 0.42-14.45 |
| Family Doctor | In-Person | 1.00 (ref) | N/A | 1.00 (ref) | N/A |
| | Remote | 1.45 * | 0.96-2.20 | 1.26 | 0.66-2.43 |
| | Hybrid | 1.55 | 0.91-2.64 | 1.92 | 0.86-4.32 |
| Psychologist | In-Person | 1.00 (ref) | N/A | 1.00 (ref) | N/A |
| | Remote | 1.43 | 0.71-2.86 | 1.76 | 0.43-7.09 |
| | Hybrid | 1.33 | 0.52-3.42 | 1.43 | 0.21-9.86 |
| Social Worker | In-Person | 1.00 (ref) | N/A | 1.00 (ref) | N/A |
| | Remote | 1.30 | 0.73-2.30 | 1.90 | 0.93-3.88 |
| | Hybrid | 2.03 | 0.93-4.42 | 0.70 | 0.22-2.27 |

* Significant at the a = 0.05 level.

^ Adjusted for Age, Gender, Visible Minority Status, and Marital Status.

the different combinations of healthcare professionals and consultation modalities for either employed or unemployed individuals.

For all models, VIF values ranged from approximately 1.00 to 1.38, indicating no concerns over collinearity.

## Discussion

Our study examined the relationship between consultation modality (in-person, remote, and hybrid) and provider type (family doctor, psychiatrist, psychologist, and social worker) on self-reported quality of care and perceived mental health outcomes, comparing employed and unemployed individuals. The results of this study revealed that, while service use patterns were generally consistent between employed and unemployed individuals, there were no statistically significant differences in perceived mental health outcomes across the consultation modality and mental health professionals for both employed and unemployed groups. However, important differences emerged in self-reported quality of care.

Both employed and unemployed participants most frequently sought help from family doctors. This is consistent with the literature, which highlights that family physicians are the first line of support for mental health problems. For instance, a study found that many individuals received mental health care either only from primary care providers or in combination with mental health professionals [17]. More people seek family physicians for mental health services than psychiatrists, social workers, psychologists, and other health professionals, further supporting our study's findings [31]. Although past literature has demonstrated this pattern, the potential influence of employment status on provider choice has not been examined. Our results contribute to the literature by demonstrating that this pattern is observed among both employed and unemployed individuals, suggesting that family doctors are perceived as a universal first point of contact, potentially due to their accessibility and lower stigma [32]. Although there were no significant differences in the perceived mental health outcomes between the two groups, these service use patterns may help explain some of the variations observed in self-reported quality of care in our study.

Our study found that among unemployed individuals, hybrid consultations, particularly when delivered by psychiatrists or family doctors, were associated with significantly higher perceived helpfulness compared to in-person care. This finding aligns with Donabedian's perspective that good structure supports effective processes, which in turn lead to better outcomes [19]. In this case, access to appropriate professionals, psychiatrists and family doctors (structure), combined with hybrid modality (process), enhanced patients' perceived quality of care (outcome). Therefore, quality of care may be influenced not only by the process (type of consultation modality) but also by the interaction between structure (type of professional) and process. Both psychiatrists and family doctors play a critical role in mental health support, providing specialized services and serving as the primary point of contact, respectively. Therefore, when offered in a hybrid format, their services may have improved both continuity and accessibility of care. This pattern was not observed with other types of professionals, such as psychologists and social workers, suggesting that psychiatrists and family doctors may be well-suited or preferred by these individuals to deliver hybrid care. These findings suggest that hybrid modalities may be more beneficial for unemployed individuals, possibly due to reduced structural barriers, such as transportation or financial limitations [16]. This aligns with Andersen's framework, which highlights enabling factors as resources that allow people to use services [33]. In this context, employment status can be considered an enabling resource as it affects income and access to services, providing an explanation as to why unemployed individuals benefitted from hybrid modalities. This aligns with the literature, which also supports this by showing that hybrid models can combine the advantages of both modalities, providing the flexibility and convenience that several unemployed individuals require, while maintaining the personal connection, physical attendance, and reduced loneliness of in-person visits [34].

In contrast, our study also found that among those who were employed, the same was insignificant, implying that there were no significant differences in the perceived quality of care between hybrid and in-person modalities across different professionals in this group. This suggests that hybrid modality did not seem to influence their perception. This may be because access to in-person care was already made possible for them due to already lower structural barriers, such as

reliable transportation and stable schedules. Another explanation could be that unemployed individuals experience poorer mental health at baseline than employed individuals. Several studies have shown that unemployed individuals experience poorer mental health outcomes than employed individuals, potentially due to financial difficulties and job loss [15,16,35]. This group may, as a result, have more needs than those who are employed, making any form of accessible care, including hybrid services, feel more beneficial.

Furthermore, literature indicates that telemental health appointments are less likely to be missed or cancelled than in-person ones, which improves continuity and effectiveness of care [36]. One of the potential reasons for this could be the effort required to seek a psychiatrist, psychologist or social worker physically. It also explains why both employed and unemployed individuals in our study, who sought social work or psychiatric services, commonly preferred remote options. Although remote care has several advantages, such as accessibility and convenience [37], existing literature also highlights its limitations, including technical difficulties and privacy concerns [37,38]. A qualitative study highlights some mixed findings. Some health professionals noted that clients were able to easily express themselves remotely from home [39]. However, others have mentioned that for some individuals, potential external factors, such as having children or a spouse at home, may restrict their ability to speak freely [39]. These findings align with our study's results, which show that hybrid consultations improve the quality of care, especially for individuals facing barriers to accessing in-person care and/or issues with fully remote care. The hybrid model offers a more flexible structure that better suits the needs of individuals, particularly those who are unemployed, and may help alleviate some of the challenges associated with fully remote care.

Interestingly, in our study, hybrid care was the least utilized model across all types of professionals in both groups, despite being associated with a higher perceived quality of care. One possible explanation for this inconsistency may be that providers are hesitant to utilize hybrid models for their patients due to uncertainty regarding rules and policies, which makes it more difficult to continue using telemental health after the pandemic [37]. The implementation of hybrid modalities may be discouraged by this uncertainty. Additionally, due to the underutilization of this modality, individuals may be unaware of its type and how it works.

However, our study found that among employed and unemployed individuals, there were no statistically significant differences in perceived mental health outcomes across consultation modality and mental health professionals. Given that both groups most frequently sought family doctors for support, this result was somewhat expected. This implies that the consistency in provider type may contribute to the similarity in perceived mental health outcomes between the two groups, suggesting that similar results are seen when family doctors are consulted. Other types of professionals, such as psychiatrists, psychologists or social workers, were consulted less frequently, which may have limited the ability to detect outcome differences across modalities and providers. However, previous literature suggests that such differences do exist, highlighting the inconsistency with our findings. For instance, compared to mental health specialists, family doctors may be less qualified to provide therapeutic support remotely. The effectiveness of these services may be impacted since generally physicians have expressed their dissatisfaction with the quality of mental health they provide and mentioned their inability to provide adequate preventive mental health services [40,41].

Conversely, some studies have indicated that consulting family doctors resulted in greater improvements in person. For instance, individuals receiving treatment from mental health specialists experienced greater improvement compared to those receiving treatment from primary care providers only [17]. However, it is essential to note that this study, which was conducted in the United States, focused on ethnic and racial minorities and therefore these minorities may experience outcomes differently when they consult different professionals. For instance, the same study also found that Hispanics who received mental health treatment from primary care providers, which included internal medicine, family medicine and general practice, experienced greater improvements, reflecting better mental health status [17]. Another potential explanation for these positive findings in the literature, aside from being a minority, relates to the type of mental health problem and its' severity [17,42].

Previous studies have noted that individuals with milder symptoms or mental health problems, like panic disorders, are more likely to seek care from family doctors, while those with more severe or persistent issues or mental health problems,

such as major depression and psychological distress, were more likely to consult with mental health professionals or different professionals [17,42]. Therefore, individuals experiencing milder symptoms may perceive greater improvements in their mental health outcomes, which explains why consulting family doctors may have led to better outcomes in some studies in the literature and makes them the first point of contact in certain situations [17]. In contrast, individuals with more persistent or severe conditions may experience smaller improvements, even when appropriate treatment is implemented [17]. While our study did not assess the severity or specific type of mental health problem, the frequent use of family doctors by both groups in our sample may reflect a greater proportion of those experiencing mild symptoms or a tendency to seek care with a family doctor first before pursuing specialized services like seeking a psychiatrist. As a result, the similar distribution of provider type and consultation modality across both employed and unemployed groups could explain the lack of significant differences in perceived mental health outcomes.

Although our study did not find significant differences in perceived quality of care or outcomes related to consultations with social workers, both employed and unemployed individuals in our sample were more likely to consult social workers remotely. There are limited findings in the literature regarding the effectiveness of social workers' services for mental health problems; however, several studies have highlighted the utilization of these services [17]. However, like family doctors, the existing literature notes that social workers may face challenges in delivering care effectively, especially when providing remote care. A qualitative study focused on a social worker's experiences highlighted the interdisciplinary and interactive nature of their work [43]. As a result, the lack of real-time collaboration with other professionals, which is common in remote settings, can negatively affect their ability to understand a patient's needs fully, ultimately influencing the quality of care and patient outcomes. The social worker also noted that in-person consultations have a positive impact on the quality of the relationship with the patient, and vice versa, ultimately leading to healing [43].

## Strengths and limitations

To our knowledge, this is the first study to examine perceived quality of care and mental health outcomes across different consultation modalities and professional types, stratified by employment status. Stratifying by employment status enabled us to identify how these outcomes vary between employed and unemployed individuals, facilitating the creation of more targeted recommendations and interventions. The study also benefited from a large dataset, particularly for the employed group, which enhanced the accuracy and precision of the odds ratios for perceived quality of care and mental health. Additionally, the sample ensures proper representation of visible minority groups due to the oversampling of these groups. The use of stratified random sampling design also allowed the findings to generally reflect the Canadian population. Finally, we assessed multicollinearity using VIF, which strengthened the robustness and validity of our analyses.

This study had several limitations that should be considered when interpreting the results, as they may have influenced the reliability of the findings. First, one form of remote consultation, text messages, and two types of mental health professionals, nurses and counsellors, were excluded due to small sample sizes. The exclusion of these variables may have limited the comprehensiveness of the findings and missed out on key insights. Another key limitation is the study's use of a cross-sectional design where causal inferences cannot be made about the impact of provider type or modality on mental health outcomes. This requires the need for future research to conduct experimental studies to confirm causality. The study could not also confirm whether all participants in the dataset had access to mental health services. This limitation is particularly important as differences in access between employed and unemployed individuals may have influenced the reported outcomes. Similarly, the data collection took place in 2022, which was a period still significantly impacted by the COVID-19 pandemic. This serves as a potential limitation as this unexpected event may have influenced both the perceived quality of care and mental health outcomes. The increased stress and uncertainty during this time may have affected how these individuals perceived their mental health, the care they received and who they sought, regardless of consultation modality and professional type. Additionally, it was the time when hybrid modalities were just starting, which could explain why hybrid options were found to be less utilized in our study. While dichotomizing the dependent variable

allows for easier comparison using odds ratios and may help reduce variability, this approach can also reduce statistical power. Lastly, the small sample sizes within subgroups of unemployed participants, specifically those seeking a psychiatrist or psychologist, resulted in wide confidence intervals and reduced precision of the odds ratios for perceived quality of care and mental health.

### Future research and implications

Although this study found no significant differences in perceived mental health outcomes between employed and unemployed individuals, notable differences in self-reported quality of care emerged across consultation modalities and provider types. These findings suggest that while mental health outcomes may appear similar across both groups, patients' experiences and satisfaction with care may differ based on who they consult and what modality was utilized. This is important as treatment adherence, future use of these services and general involvement with the mental health system can all be impacted by the perceived quality of care. These findings also demonstrate the importance of tailoring mental health interventions not only by provider type and consultation modality (e.g., remote, in-person, or hybrid), but also by individual characteristics, such as age, gender and employment status.

Future research should investigate the specific factors contributing to the differences in perceived quality of care between employed and unemployed individuals, such as accessibility and waiting times, by analyzing their experiences through a qualitative lens. Additionally, just as hybrid models that combine remote and in-person care were analyzed in this study as a variable, future research should further explore the combination of specific providers, such as collaborative care between family doctors, psychiatrists, and social workers, and see how this impacts patients' quality of care and outcomes among employed and unemployed individuals. Interprofessional collaboration may be the key to improving both perceived quality of care and outcomes among these individuals, especially when complex mental health problems are involved. Moreover, given how we had a low sample size for those categorized as unemployed, future studies should aim to recruit larger numbers of unemployed individuals to allow more precision comparisons across mental health professional types.

Furthermore, future mental health services should consider integrating occupational therapists to reduce wait times and improve access. These professionals can also provide valuable assessments of patients' functional abilities and help identify occupational barriers to recovery. Their inclusion could enhance the quality of mental health care, especially for patients with impairments that affect their work or daily living.

### Conclusion

This study examined the relationship between the type of consultation — whether in-person, remote, or hybrid — with various mental health professionals and self-reported quality of care and perceived mental health among employed and unemployed Canadians. By analyzing both the modality of care and the specific professional consulted, this research highlighted several ways in which mental health services are experienced across employment groups. Notable differences emerged in self-reported quality of care. Future research should further investigate how interprofessional collaboration can enhance the quality of care and mental health outcomes.

### Author contributions

**Conceptualization:** Aishvinigaa Sathananthan, Behdin Nowrouzi-Kia.

**Data curation:** Kishana Balakrishnar.

**Formal analysis:** Kishana Balakrishnar.

**Project administration:** Aishvinigaa Sathananthan.

**Supervision:** Behdin Nowrouzi-Kia.

**Visualization:** Kishana Balakrishnar.

**Writing – original draft:** Aishvinigaa Sathananthan, Kishana Balakrishnar, Raihana Premji.

**Writing – review & editing:** Aishvinigaa Sathananthan, Behdin Nowrouzi-Kia.

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
