## [Decision Letter · Decision Letter 0]

21 Sep 2025

PMEN-D-25-00353

The influence of consultation modality and mental health professional on perceived quality of care and mental health outcomes: A comparison between employed and unemployed individuals

PLOS Mental Health

Dear Dr. Nowrouzi-Kia,

Thank you for submitting your manuscript to PLOS Mental Health. After careful consideration, we feel that it has merit but does not fully meet PLOS Mental Health’s publication criteria as it currently stands. Therefore, we invite you to submit a revised version of the manuscript that addresses the points raised during the review process.

In addition to addressing reviewer comments, consider the following points related to section "Statistical analyses".

1. The decision to dichotomize both dependent variables (quality of care and perceived mental health) has no rationale provided. Dichotomization may lead to loss of information and statistical power.

The authors should explain why this approach was preferred over treating the variables as ordinal or continuous (e.g., using ordinal logistic regression or treating scales as ordered categories).

2. The text does not describe how model fit was assessed (e.g., multicollinearity among predictors checked, if logistic regression assumptions validated, lack of influential outliers).

This section is recommended to be improved.

We look forward to receiving your revised manuscript.

Kind regards,

Ariel Soares Teles

Academic Editor

PLOS Mental Health

Journal Requirements:

1. Please ensure that your Ethics Statement is available in its entirety at the beginning of your Methods section, under a subheading 'Ethics Statement'. It must include:

1) The name(s) of the Institutional Review Board(s) or Ethics Committee(s)

2) The approval number(s), or a statement that approval was granted by the named board(s) 

3) (for human participants/donors) - A statement that formal consent was obtained (must state whether verbal/written) OR the reason consent was not obtained (e.g. anonymity). 

NOTE: If child participants, the statement must declare that formal consent was obtained from the parent/guardian.

2. In the online submission form, you indicated that “The dataset used and/or analyzed during the current study are available upon request from the corresponding author”. 

3. Uploaded as supplementary information.

Additional Editor Comments (if provided):

In addition to addressing reviewer comments, consider the following points related to section "Statistical analyses".

1. The decision to dichotomize both dependent variables (quality of care and perceived mental health) has no rationale provided. Dichotomization may lead to loss of information and statistical power.

The authors should explain why this approach was preferred over treating the variables as ordinal or continuous (e.g., using ordinal logistic regression or treating scales as ordered categories).

2. The text does not describe how model fit was assessed (e.g., multicollinearity among predictors checked, if logistic regression assumptions validated, lack of influential outliers).

This section is recommended to be improved.

Reviewers' comments:

Reviewer's Responses to Questions

**Comments to the Author**

1. Does this manuscript meet PLOS Mental Health’s publication criteria?

Reviewer #1: Partly

Reviewer #2: Partly

2. Has the statistical analysis been performed appropriately and rigorously?

Reviewer #1: Yes

Reviewer #2: Yes

3. Have the authors made all data underlying the findings in their manuscript fully available (please refer to the Data Availability Statement at the start of the manuscript PDF file)?

Reviewer #1: Yes

Reviewer #2: No

4. Is the manuscript presented in an intelligible fashion and written in standard English?

Reviewer #1: Yes

Reviewer #2: Yes

Reviewer #1: Review report_Manuscript No.: PMEN-D-25-00353

This study presents sound findings on the relationship between consultation modalities and the types of mental health professionals consulted and perceived service quality and mental health outcomes, using a comparison between employed and unemployed individuals. I believe that these findings would not only advance knowledge in the field of mental health but also inform mental health professionals about consultation modalities that lead to greater satisfaction among service users. I therefore congratulate the authors on their excellent work. However, I made the following suggestions that could further improve the quality of the manuscript.

Title

After reading the manuscript, I understand that this study is observational and not interventional. If the authors agree with me, the word “influence” in the title could be replaced by “relationship.” Also, in the title, the word ‘type’ is missing from the concept of “mental health professional.” I therefore suggest that the title be changed to

“The relationship between consultation modality and mental health professional type and perceived quality of care and mental health outcomes: A comparison between employed and unemployed individuals”

Abstract

In this sentence (lines 28-30), I suggest saying: “The type of mental health professional consulted and the type of consultation modality may be associated with individuals’ perceptions...”

Lines 30-35, it seems to me that the study aims to examine the relationship between the type of consultation with different mental health professionals and the perceived quality of care and mental health outcomes...

Introduction

If I understand correctly, this study focuses explicitly on mental health professionals. In the context in which the study was conducted, are family doctors (also known as general practitioners in some countries) officially included on the list of mental health professionals alongside psychiatrists, psychologists, etc.? If so, I advise mentioning this explicitly and providing references. If not, please clarify their position as mental health professionals in this study or remove them from the list.

For the purpose of the study (lines 141-144), I suggest stating that: This study aims to examine the relationship between the type of consultation (...) with different types of mental health professionals (...) and the quality of care and self-reported mental health among employed and unemployed Canadians.

Methods

The study design does not seem very clear. Please clarify it a little.

Some words (e.g., Black, Chinese, Filipino, South Asian) are repeated (see Lines 157-158). Please check if this is an error.

Results

The results are very well presented. However, in Table 4, although no relationship between variables has been established, the confidence intervals for the two types of mental health professionals (psychiatrists and psychologists) appear to be very wide in the “unemployed” category. Is there a possible explanation for this?

Discussion

In the following statement: “Our study examined the impact of consultation modality (in-person, remote and hybrid) and provider type (family doctor, psychiatrist, psychologist, and social worker) on self-reported quality of care and perceived mental health outcomes, comparing employed and unemployed individuals.” (Lines 252-255), I would suggest deleting the word “impact” and rewriting it as follows: “Our study examined the relationship between consultation modality (...) and provider type (...) and self-reported quality of care and perceived mental health outcomes, comparing employed and unemployed individuals.”

The limitations of the study are very well elaborated. One limitation that could also be discussed is the type of study design used, which does not allow for establishing cause–effect relationships. However, the study strengths are not described. I suggest titling this subsection “Strengths and Limitations” and elaborating on the methodological study strengths.

In the conclusion (lines 424-426), I would suggest stating that: “This study examined the relationship between the type of consultation—whether in-person, remote, or hybrid—with various mental health professionals and self-reported quality of care and perceived mental health among employed and unemployed Canadians.”

References

In the body of the manuscript, please harmonize the citation style for references. Two citation styles are currently used: the numerical style [1], [2], [5] and the author-date style: Snoswell et al. (2021), Henry et al. (2020), Li and Nowrouzi-Kia (2024), Jenkins-Guarnieri et al. (2015).

Thank you.

Reviewer #2: Need of the review and the methodology adopted in this review should be stated with supporting literatures. A theoretical framework should be added in this paper. The added information would make the paper more clear and scientific for the authors. I recommend this paper for publication with minor edits.

**Do you want your identity to be public for this peer review?** For information about this choice, including consent withdrawal, please see our Privacy Policy

Reviewer #1: **Yes: ** Erick Mukala Mayoyo

Reviewer #2: No

---

## [Decision Letter · Decision Letter 1]

20 Oct 2025

The relationship between consultation modality and mental health professional type and perceived quality of care and mental health outcomes: A comparison between employed and unemployed individuals

PMEN-D-25-00353R1

Dear Dr. Nowrouzi-Kia,

We are pleased to inform you that your manuscript 'The relationship between consultation modality and mental health professional type and perceived quality of care and mental health outcomes: A comparison between employed and unemployed individuals' has been provisionally accepted for publication in PLOS Mental Health.

Best regards,

Ariel Soares Teles

Academic Editor

PLOS Mental Health

Reviewer Comments (if any, and for reference):

Reviewer's Responses to Questions

**Comments to the Author**

Reviewer #1: All comments have been addressed

publication criteria?

Reviewer #1: Yes

3. Has the statistical analysis been performed appropriately and rigorously?

Reviewer #1: Yes

4. Have the authors made all data underlying the findings in their manuscript fully available (please refer to the Data Availability Statement at the start of the manuscript PDF file)?

Reviewer #1: Yes

5. Is the manuscript presented in an intelligible fashion and written in standard English?

Reviewer #1: Yes

Reviewer #1: Thank you for addressing all my concerns. I have no further comments. I hope this has helped to improve the quality of the manuscript, which I believe will contribute to the advancement of knowledge in the field of mental health.

**Do you want your identity to be public for this peer review?** For information about this choice, including consent withdrawal, please see our Privacy Policy

Reviewer #1: **Yes: ** Erick Mukala Mayoyo
